# Leaf-Level Spectral Fluorescence Measurements: Comparing Methodologies for Broadleaves and Needles

**Paulina A. Rajewicz [1],\*, Jon Atherton [1], Luis Alonso [2]** 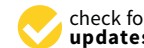 **and Albert Porcar-Castell [1]**

[1]  Optics of Photosynthesis Laboratory, Institute for Atmospheric and Earth System Research/Forest Sciences, Faculty of Agriculture and Forestry, University of Helsinki, 00014 Helsinki, Finland; jon.atherton@helsinki.fi (J.A.); joan.porcar@helsinki.fi (A.P.-C.)

[2]  Image Processing Laboratory (IPL), University of Valencia, Paterna, 46980 València, Spain; luis.alonso@uv.es

\*  Correspondence: paulina.rajewicz@helsinki.fi; Tel.: +358-50318-6952

**Abstract:** Successful measurements of chlorophyll fluorescence (ChlF) spectral properties (typically in the wavelength range of 650–850 nm) across plant species, environmental conditions, and stress levels are a first step towards establishing a quantitative link between solar-induced chlorophyll fluorescence (SIF), which can only be measured at discrete ChlF spectral bands, and photosynthetic functionality. Despite its importance and significance, the various methodologies for the estimation of leaf-level ChlF spectral properties have not yet been compared, especially when applied to leaves with complex morphology, such as needles. Here we present, to the best of our knowledge, a first comparison of protocols for measuring leaf-level ChlF spectra: a custom-made system designed to measure ChlF spectra at ambient and 77 K temperatures (optical chamber, OC), the widely used FluoWat leaf clip (FW), and an integrating sphere setup (IS). We test the three methods under low-light conditions, across two broadleaf species and one needle-like species. For the conifer, we characterize the effect of needle arrangements: one needle, three needles, and needle mats with as little gap fraction as technically possible. We also introduce a simple baseline correction method to account for non-fluorescence-related contributions to spectral measurements. Baseline correction was found especially useful in recovering the spectra nearby the filter cut-off. Results show that the shape of the leaf-level ChlF spectra remained largely unaffected by the measurement methodology and geometry in OC and FW methods. Substantially smaller red/far-red ratios were observed in the IS method. The comparison of needle arrangements indicated that needle mats could be a practical solution to investigate temporal changes in ChlF spectra of needle-like leaves as they produced more reproducible results and higher signals.

**Keywords:** baseline correction; chlorophyll fluorescence; FluoWat; leaf morphology; lingonberry; photosynthesis; Scots pine; silver birch; sun-induced fluorescence

## 1. Introduction

Solar-induced fluorescence (SIF) can be remotely measured from towers, unmanned aerial vehicles (UAVs), aircrafts, and satellites [1–4]. SIF has shown strong correlation with gross primary productivity (GPP) [4–7]. This relationship can potentially be used to improve the accuracy of global carbon cycle assessments and to refine dynamic vegetation models via data assimilation schemes. However, SIF data can only be retrieved within discrete and narrow Fraunhofer or atmospheric absorption bands [8,9], while chlorophyll fluorescence (ChlF) is in fact emitted as a continuous spectrum from (approx.) 650–850 nm [3]. Characterization of the factors that control the variation in shape and amplitude of the ChlF spectra, starting from the leaf level, is therefore a cornerstone for the interpretation of

remotely-sensed SIF [10,11]. Fortunately, the complete spectrum is measurable at the leaf-level, which facilitates the investigation of the connection between different ChlF spectral bands and physiological leaf functionality.

At the leaf scale, the shape of the ChlF spectra is characterized mainly by two peaks, located around 685–690 (red peak) and 740–750 nm (far-red peak) [12–14]. Variations in peak amplitude can affect the red/far-red ratio (i.e., F685/F740) and depend on: (a) the dynamic response of leaves to environmental and stress factors (e.g., to temperature [15] or water stress [16]); (b) light absorption and scattering effects within the leaf, associated with variation in leaf-level chlorophyll concentration [Chl] [17,18]; and (c) the architecture of the photosynthetic apparatus and the relative contributions of photosystem II (PSII) and photosystem I (PSI) to the ChlF emission [11,19–21]. At ambient temperature it is, however, challenging to ascribe specific emission bands to populations of PSI and PSII as there is a large degree of spectral overlap between the peaks [19,22,23]. Interestingly, the ChlF emission of the two photosystems can be separated by measuring ChlF spectra at 77 Kelvin (77 K) [15,24,25], where the red ChlF wavelengths correspond to PSII and the far-red bands to PSI.

Typically, protocols for spectral measurements are based on setups that use a light source (artificial or the Sun) coupled with a short-pass filter that removes radiation above a given threshold (e.g., 650 nm). The setups then include a chamber or some other means to hold a sample at a given geometry with respect to the light source and spectrometer fiber. Irrespective of the selected methodology, it is a particular challenge to measure small and narrow leaves of complicated geometries, like those of conifers, in comparison to the more easily measured, relatively wider and flatter broadleaves.

Nonetheless, conifers represent a significant fraction of terrestrial ecosystems—in 2010, 39% of the world's total growing stock [26]. It is, therefore, important to face the technical challenge of measuring needles with varying geometrical shapes and thicknesses. Measuring a single needle inevitably leads to a diminution of the signal-to-noise ratio (SNR) [27]. At the same time, measuring more than one needle creates a need for arranging needles into mats (e.g., [27]) or for using needle holders (e.g., [28]). These two-dimensional configurations can, in turn, enhance mutual shading of adjacent needles, multiple scattering, or re-absorption [29].

Several studies [27,30–33] have assessed the optimal arrangement of needles for optical measurements of reflectance (R) and transmittance (T). Due to the design of the most prevalent instrumentation (integrating spheres with port sizes on the order of cm) these studies typically measured multiple needles simultaneously. Inevitably, there are gaps between needles in such a measurement; the gap fraction ($G_F$) is, therefore, a parameter of considerable importance when conducting such measurements. Small $G_F$ have been shown to yield unrealistic negative T values for the photosynthetically active radiation (PAR) region [32,33]. Mesarch et al. stated that errors are larger in samples with large $G_F$ while compared to small $G_F$ and that large $G_F$ affects T more negatively than R [34]. At the same time, Yáñez-Rausell suggested that effect of the needles' own cross-section had a stronger negative effect on measurements accuracy than the $G_F$ itself [35]. Hence, gaps are both inevitable and a source of considerable error when measuring the spectral properties of needle-like leaves.

As far as we are aware, there have been no previously reported efforts to characterize the influence of gap fraction on measurements of spectral ChlF. When considering such a study it is important to distinguish the nature of different optical processes measurements. Gaps between adjacent needles will promote multiple scattering of photons, which can bias R and T measurements. In case of ChlF, on top of multiple scattering, one also has to consider the re-absorption of scattered red ChlF by adjacent needles and subsequent enrichment in far-red ChlF. Accordingly, gaps could potentially affect both the magnitude and shape of the ChlF spectra. Therefore, while some technical challenges of needle measurements in general can be concluded, R and T studies should not lead to assumptions on how sample morphology or arrangement affect ChlF. As this effect remains unknown, standardization of

needle- and leaf-level measurements is key to agreeing on accurate interpretation of all-levels ChlF from various species.

The aim of this study was to evaluate the functionality of different methods and discuss their applicability in different practical conditions. Leaf-level ChlF has been measured with various methods, also using FluoWat (e.g., [18,36]). However, a question whether or how the geometry of measurements (viewing and illumination angles) affects the measured ChlF spectral shape across leaf samples with different morphology or across needle samples of different arrangements, was not yet solved in the literature.

Here we compare three methods: (1) A custom made optical chamber designed to measure ChlF of leaf samples at both ambient and 77 K temperatures in the backscatter direction; (2) FluoWat, a widely used clip originally designed for in situ measurements of SIF spectra, which can measure both up-welling and down-welling ChlF at a 45° angle respective to the illumination beam; and (3) an integrating sphere to estimate total hemispherical ChlF emission spectra of samples illuminated at nadir. In addition, we compare measurements of broadleaves and needle-leaves and assess the impact that different needle arrangements have on the resulting ChlF spectral properties. A first approximation protocol to recover the red ChlF tail around the cut-off filter and to correct for potential baseline effects is also introduced.

## 2. Materials and Methods

### 2.1. Plant Material and Experimental Protocol

Leaves from three different species growing in the vicinity of the University of Helsinki, Viikki Campus, Finland were collected towards the end of the growing season (September 2017): silver birch (*Betula pendula L.*), a deciduous tree with relatively thin leaves (specific leaf area, SLA = 161 $cm^2g\,DW^{-1}$); lingonberry (*Vaccinium vitis-idaea* L.), an evergreen shrub with relatively thick leaves (SLA = 55 $cm^2g\,DW^{-1}$); and Scots pine (*Pinus silvestris* L.), an evergreen tree with needle-leaves (SLA = 39 $cm^2g\,DW^{-1}$). One exposed and south-facing branch from the accessible part of the canopy per each tree was cut during early morning and rapidly brought to the lab. For lingonberry, three nearby shoots where cut to provide enough leaf material. All shoots were immediately re-cut under and kept in water, in the dark, and at room temperature through the duration of measurements (appox. eight hours). Species and replicates (15 measuring points: three species × five replicates) were randomized within each method. Methods were not randomized due to practical reasons. Leaf-level reflectance and spectral ChlF were measured with three different methods: (1) optical chamber (OC); (2) FluoWat (FW); and (3) integrating sphere (IS). A fresh leaf or set of needles was always used at each measuring point. Measurements with OC started 30 min after re-cutting branches under water, dark acclimation and temperature stabilization in the lab. Measurements with FW and IS were taken approx. 90 and 240 min after re-cutting branches, respectively.

The size of the birch and lingonberry leaves was sufficient to cover the measuring area in OC and FW as well as sample-port in the IS. In contrast, different configurations were possible for conifer measurements. We tested three different configurations (Figure 1): (a) single needle (1N); (b) three needles with an approximately one-needle-wide gap in between each (3N); and (c) continuous needle mat with as minimal gaps as technically possible (NM). Measurements were conducted with a background consisting of a photon trap (FW and IS) or a non-fluorescing black tape (see Supplementary Material, Figure S1) in OC.

Light sources were adjusted for each method to yield similar low light intensities at the leaf surface (as measured with an LI-250A PAR sensor, LI-COR®Inc., Lincoln, NE, USA) (Table 1). Despite the obvious decrease in SNR, we purposefully selected a low light intensity (38–50 μmol PAR) to minimize any potential differences in ChlF spectral properties due to light-induced photochemical quenching (PQ) and non-photochemical quenching (NPQ) dynamics [11]. To ensure we were comparing ChlF spectra from fully functional leaves of similar physiological states, we calculated the maximum quantum yield of photochemistry based on the ratio of variable to maximal fluorescence, or $F_V/F_M$ [37],

and used it as an indicator of photosynthetic functionality [24]. We used a PAM-2500 fluorometer (Heinz Walz GmbH, Effeltrich, Germany) and the cut shoots after one hour of dark acclimation (n = 5). $F_V/F_M$ remained at the level of 0.78 ± 0.023 for lingonberry, 0.79 ± 0.012 for birch, and 0.83 ± 0.012 for pine, which is well within the range of non-stressed summer values for each of the respective species [38,39]. More details can be found in Table 1 and the sections below.

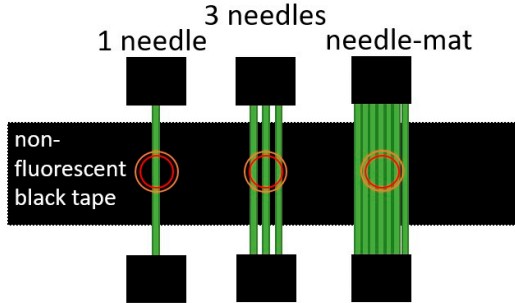

**Figure 1.** Illustration of the three different needle configurations used during measurements in OC. Analogous configurations were used in FW but using a light trap as background. Circles symbolize an approximate field of view in OC (red) and FW (orange).

**Table 1.** Key elements in the optical chamber, FluoWat, and integrating sphere setups. See Section 2.2. for further details on each method.

| Feature | Optical Chamber | FluoWat | Integrating Sphere |
|---|---|---|---|
| Light source | Ocean Optics®HL-2000 (halogen) | | ASD®RTS-3ZC dedicated light source (halogen) |
| Light intensity at leaf surface | 50 µmol | | 38 µmol |
| Spectrometer | Ocean Optics®USB2000+ (range: 200–1100 nm, FWHM (600–800 nm): 1.5–1.8 nm) | | |
| Optical fiber | Ocean Optics®R600-7-VIS-125F (600 µm) | ASD (600 µm) | |
| Background | black tape | FW light trap | IS light trap |
| Filter | Thorlabs®(cut-off wavelength 650 nm, optical density = 4) | Edmund Optics®(cut-off wavelength 650 nm, optical density = 4) | |

All the measurements were acquired with an Ocean Optics®USB2000+ (Ocean Optics®, Largo, FL, USA) spectrometer and recorded with SpectraSuite (Ocean Optics®) software. All references (REFs) were collected using a reflectance standard panel (Spectralon®Diffuse Reflectance Standards, Labsphere®, North Sutton, NH, USA). Dark current (DC) and reference measurements were collected before and after each sample measurement, for DC correction and to check for stability during measurements (Figure 2). In addition, one reference measurement (REF3 in Figure 2) with the filter and at the long integration time (IT) was recorded for each method, to be used in baseline calculations (see *2.5. Baseline* for details of using REF3).

Integration times (Table 2) were optimized for different purposes: short IT was optimized for each method so that the reference spectra recorded with the Spectralon®panel would peak at approximately 90% of the maximum dynamic range of the spectrometer. Since the distance between sample and fiber and measurement geometries differed substantially within methods, these IT were very different. Similarly, a long IT was selected for each method to record a strong ChlF signal with enough amplitude, which was especially weak due to our requirement of low light illumination. For the IS method, a

longer IT would have been needed but 60 seconds was the maximum IT allowed by the spectrometer. Using the long IT came at the expense of signal saturation at wavelengths lower than the cut-off filter.

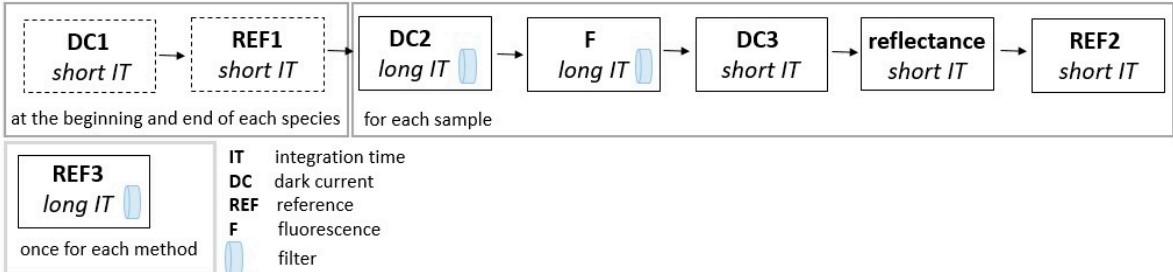

**Figure 2.** Flux diagram of the experimental protocol used for acquiring leaf ChlF and reflectance (R) spectra. Note the combination of long and short integration times (IT). See Section 2.2. for further details on each method.

**Table 2.** Integration times and spectral averaging used in the optical chamber, Fluowat, and integrating sphere protocols.

|  | Integration Time | Optical Chamber | FluoWat | Integrating Sphere |
|---|---|---|---|---|
| *short* | Integration time (ms) | 14 | 100 | 1000 |
|  | Number of averaged spectra | 25 | 10 | 10 |
| *long* | Integration time (ms) | 300 | 15,000 | 60,000 |
|  | Number of averaged spectra | 1 | 1 | 1 |

Although we acknowledge that signal saturation is undesirable and generally not recommended by the manufacturer, it may sometimes be the only solution to record a spectral feature with a very weak signal relative to other regions of the spectra (as in the present case), especially when using spectrometers with a limited dynamic range, or under situations where the user is interested in the temporal information of the signal (i.e., spectral averaging is not desired). The visible effect of using a long IT and saturating the signal is the appearance of a substantial baseline that needs to be corrected (see Section 2.3).

## 2.2. Description of Methods

### 2.2.1. Optical Chamber

The optical chamber (OC) is a customized setup originally developed to meet three requirements: (i) facilitate the arrangement of different types of leaf samples in the cuvette, where samples can be carefully attached with the help of magnets or tape; (ii) tightly accommodate an off-the-shelf reflectance probe fiber holder (see details below), so that it can be removed and placed again to measure exactly the same leaf area; and (iii) allow measuring both ambient and liquid nitrogen temperature (77 K) ChlF spectra of the same sample area, where liquid nitrogen is poured into the chamber to reduce leaf temperature to 77 K. The setup can be used for measuring directional reflectance and ChlF of leaves in the backscatter direction both at nadir (with specular reflectance component but enhanced ChlF signal), and at 45° (no specular reflectance but slightly smaller ChlF signal). We here selected the nadir arrangement to add further differentiation from the FluoWat method. Besides the elements listed in Table 1, the setup (Figure 3a) consisted of a 3-D printed PLA plastic cuvette (see Supplementary Material, Figure S2 for cuvette design). A series of detachable metal plates were used to facilitate the arrangement of multiple samples, the use of magnets, and for added chamber protection at 77 K. A fiber holder (Ocean Optics®, RPH-1) was used to keep the bifurcated reflectance probe (OceanOptics®R600-7-VIS-125F) at a constant position in the chamber. A multimode round fiber optic bundle cable (BF20HSMA, ThorLabs®, Newton, NJ, USA) connected the light source to the filter

carrier (OceanOptics®FHS-UV). Samples (leaf or needles) were arranged and fixed on the metal plates using a non-fluorescent black tape (see Figure 1 and Supplementary Material, Figure S1). Plates with leaf samples were then dark acclimated for approx. 30 min.

### 2.2.2. FluoWat

Developed at the University of Valencia, FluoWat (FW) is a customized, portable leaf clip designed to measure optical leaf traits either in the field (using the Sun as a light source) or in the lab [36,40]. The FW clip can be used to measure directional reflectance (R) and transmittance (T) factors, as well as top and underside leaf-surface ChlF spectra. In our study, only top leaf surface R and ChlF spectra were measured. The clip design is intended to illuminate the leaf surface at a 45° angle and measure reflectance, transmittance and ChlF spectra at a nadir view (Figure 3b). For ChlF measurements, a short-pass filter (Edmund Optics®, Barrington, NJ, USA) with optical density equal to 4 (see Supplementary Material, Figure S3) was used to exclude > 99.99% of the radiation above the cut-off 650 nm wavelength [36]. For illumination, we used the same light source and fiber optic bundle as in OC. The fiber bundle was mounted in a stand and adjusted to the selected PAR level at the leaf surface (Table 1). Reflected radiation and ChlF were transmitted via a 600 μm fiber optics (ASD Inc., Boulder, CO, USA).

### 2.2.3. Integrating Sphere

A three-inch integrating sphere (ASD RTS-3ZC, ASD Inc., Boulder, CO, USA) was used to measure directional-hemispherical reflectance and ChlF spectra of leaf samples. We included the IS into the study with the attempt of comparing the shape of the directional ChlF spectra as recorded with OC and FW with that recorded by integrating all directions (hemispherical) inside the sphere. The sphere comes with a dedicated light source, equipped with a halogen light (10 W, 6 V, Model 64225, Osram, Munich, Germany), collimator lens, and a filter carrier. For reflectance/reference measurements (see Figure 2) collimated light was provided to the reference panel or leaf sample in the nadir direction (Figure 3c). The same fiber optics as for FW was connected to the sphere's north pole port to transmit radiation to the spectrometer. For ChlF measurements, the 650 nm cut-off filter was placed in the filter carrier in front of the light source.

### *2.3. Baseline Correction*

Since only a very small 1–2% of absorbed light is usually re-emitted as ChlF in the photosystems, and due to strong re-absorption of ChlF at the level of the thylakoid membrane as well as within the leaf [11], the resulting ChlF signal emitted by a leaf can be as low as three orders of magnitude smaller than the incoming radiation. As a result, when measuring the ChlF spectra of leaves with a low-pass filter and spectrometer (as described above), we often face the situation where the digital counts for the measured spectra are close to saturation for the spectral region below the cut-off filter (in this case below 650 nm), while the ChlF signal is about two orders of magnitude smaller (or even less) and dangerously close to the level of background noise. Although this situation can be partly avoided by using a spectrometer with a high dynamic range and high SNR or by applying a strong averaging, this is not always an option. Alternatively, if reflectance dynamics are not of interest, the integration time can be increased to enhance SNR across the ChlF spectral range at the expense of signal saturation before the cut-off filter. Signal saturation *per se* can, however, affect other regions of the spectra by generating an additional component of "dark (or rather saturation-induced) current" in the detector, which may require correction (I).

A second challenge associated with ChlF spectral measurement is that of the filter transmittance. This can be relevant when using halogen light sources (which have a high output in the red and especially near-infrared (NIR) regions) compared to RGB or white LED light sources which emit mainly in the visible light regions, and especially when using cut-off filters with a low optical density (OD). Since the measured ChlF emission can be as low as three orders of magnitude smaller than the

incoming light, a filter with OD = 3 or even OD = 4 (transmission of 0.001 or 0.0001, respectively) could potentially let through enough photons to interfere with the measured ChlF. This can be especially relevant for recovering the red ChlF tail, typically near the cut-off region, where filter transmittance can still be significant. The two filters used in this study were labeled OD = 4, although, according to manufacturer specifications, transmittance was well below OD = 5 for most of the exclusion region (see Supplementary Material, Figure S3). We, therefore, assumed that filter effects can be considered negligible for an OD = 4 filter and would only require correction (II) around the cut-off zone (650–660 nm).

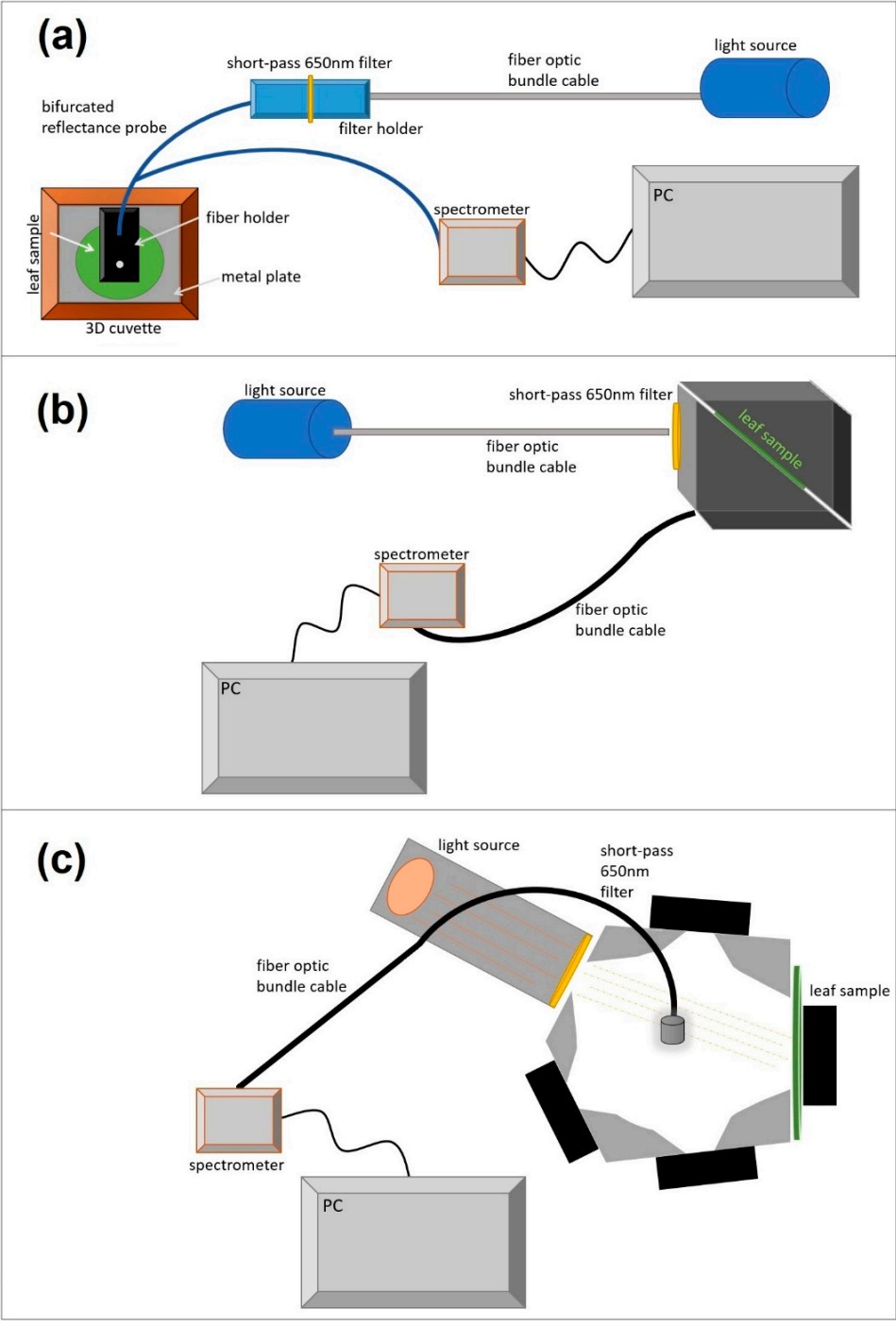

**Figure 3.** Schematic of the three different setups: (**a**) optical chamber; (**b**) FluoWat; and (**c**) integrating sphere.

A third challenge is related to the spectral resolution of the spectrometer, which we expected to interact with the measured ChlF spectra especially around the red tail. Sharp spectral features, like the line spectra of atomic emission bands used in wavelength calibration lamps, cannot be resolved as lines with spectrometers, but rather follow a normal probability distribution centered on the line's emission peak. The width of this distribution at half the maximum height, called full-width half maximum (FWHM), denotes the spectral resolution of a spectrometer. Accordingly, depending on the spectrometer resolution, light intensity, and integration time, any spectral feature will be visible in terms of spectrometer counts for a variable range of wavelengths around it. A similar phenomenon can operate when using a cut-off filter that rapidly reduces the intensity of radiation reaching the detector within a narrow cut-off region (e.g., from 74% transmittance at 644 nm to less than 1% at 654 nm; see Supplementary Material, Figure S3). We suggest that this limitation might also interfere with the measured ChlF spectra, especially when long ITs are saturating the spectral region prior to the cut-off filter, requiring correction (III).

Overall, the three phenomena described above may generate a significant and wavelength-dependent baseline level that can affect not only the intensity but also the shape of the measured ChlF spectra. We here present and implement a preliminary and simple correction method to correct for effects I, II, and III described above.

We first measure the extent of the baseline using a Spectralon®panel, with the filter in place and at the same IT as during measurements (REF3, see Figure 2). This baseline, without a leaf, will include all three effects above (see Figure 4, grey dotted line). However, because the baseline we want to correct should correspond to the situation when a sample is being measured, REF3 cannot be directly applied as a baseline. If we assume that the combined effect of I and II depends on the amount of energy reaching the detector, then we would expect the baseline for a sample $i$ to be smaller than REF3 by a factor equal to the average leaf reflectance of a sample $i$ ($\overline{\rho_{AB}}$) for a region between wavelengths A and B prior to the cut-off filter (we used here 550–650 nm, although other ranges showed comparable results), so that:

$$Baseline_i(\lambda) = REF3\,(\lambda) \quad \times \quad \overline{\rho_{i_{AB}}} \tag{1}$$

Correcting for the filter effect (II) would require precise knowledge of the spectral filter transmission $T_F(\lambda)$, for that particular filter and conditions, of the reference spectra without filter ($REF1(\lambda)$), and of the reflectance spectra of that particular leaf sample $\rho_i(\lambda)$. The leaf reflectance $\rho_i(\lambda)$ spectra needs to be multiplied because the photons potentially passing through the filter and reaching the detector will depend on the reflectance of the sample.

$$Baseline_{i\_filter}(\lambda) = REF1(\lambda) \quad \times \quad T_F(\lambda) \quad \times \quad \rho_i(\lambda) \tag{2}$$

The reflectance term is important because in case of significant filter transmission, it could add substantial wavelength-dependent artefacts into the measured ChlF spectra as a result of the red edge feature in the reflectance of green leaves. Note also that in the absence of effects I and II above, $REF1(\lambda) \quad \times \quad T_F(\lambda)$ in Equation (2) should correspond to $REF3(\lambda)$ in Equation (1). We originally tried to correct our data using Equation (2), but it resulted in an overcorrection of the ChlF spectra, indicating that effects I and II above are probably dominating the baseline (data not shown). Finally, since the manufacturer transmittance data for the filters used herein indicated that transmittance 10 nm above the cut-off region was negligible (OD > 4) (see Supplementary Material, Figure S3), and since leaf reflectance in the 650–660 nm region can be assumed to be relatively similar to the $\overline{\rho_{i_{AB}}}$ in Equation (1), Equation (2) becomes a good approximation of Equation (1) for the wavelength range where this correction may have an impact. As a result, we choose to use Equation (1) to calculate the baseline correction for each sample and, as a first approximation, assume it accounts for the integrated effect of all three factors (see Figure 4). It is important to note that this assumption will not hold if the filter transmittance is substantially higher, in which case filter transmission would need separate correction.

To further test the effect of potential filter transmittance on the resulting ChlF spectra and validity of the correction method, we carried out additional ChlF spectra measurements of birch leaves following the FW protocol. Two different light sources were tested: the halogen light described above (which strongly emits in the ChlF wavelengths) and a powerful LED light source with very small emission in the NIR (MagicShine, MS-602, Shenzhen, China), which was attenuated using a neutral density filter (OD = 1, ThorLabs ®NE510B) so that PAR at the leaf surface would be similar between light sources (see Supplementary Material, Figure S4c).

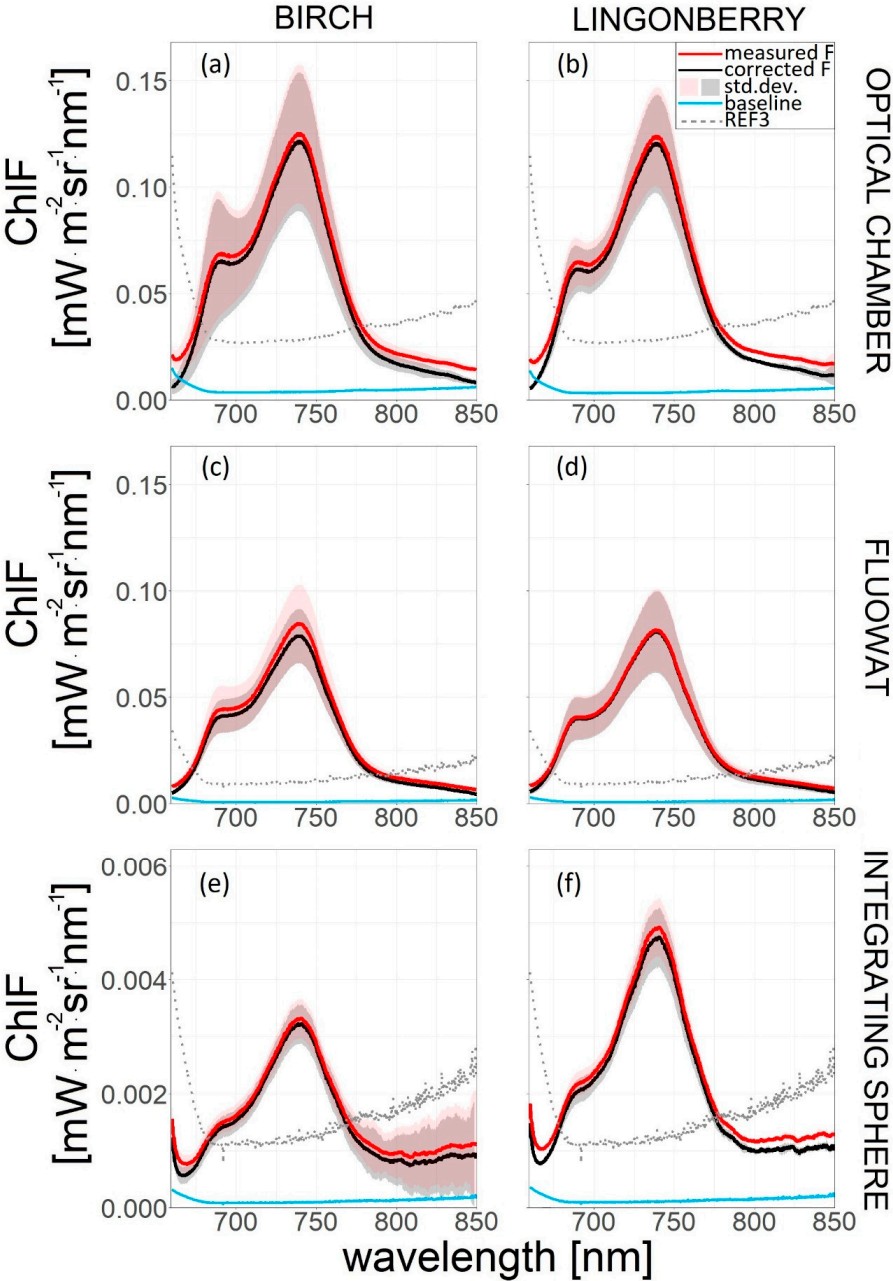

**Figure 4.** Examples of baseline correction in birch (**a**,**c**,**e**) and lingonberry (**b**,**d**,**f**) leaves for the three methods: OC (**a**,**b**), FW (**c**,**e**) and IS (**e**,**f**). Measured ChlF spectra (red line), reference measurement with Spectralon®panel and filter at the same integration time (REF3, grey dotted line), baseline estimated via Equation (1) (blue line), and corrected ChlF spectra (black line) are presented. Standard deviations marked as shadings. Note differences in y-axis scales between methods.

## 2.4. Data Analysis and Presentation

We estimated normalized ChlF spectra in an attempt to analyze changes in the shape of the ChlF spectra. We present our results as the averaged value of 4 or 5 replicates per each species. Cases of four replicates are due to errors in measurement protocol. We decided to normalize the spectra by the value obtained at the far-red peak (F740) so that the normalized value around the red wavelengths would provide a direct indication of the widely used red/far-red ChlF ratio. We subsequenty used the normalized ChlF values at 685nm (corresponding to red/far-red ratio) as the indicator to test for differences in ChlF shape between measuring protocols and arrangements. Given the small sample size (n = 4–5) a non-parametric Kruskal–Wallis test, combined with a kuskamc function for post-hoc analysis, was implemented in RStudio (vs. 3.3.4. Boston, MA, USA) and used to compare differences in red/far-red ratio between setups and arrangements. We used standard deviation (SD) of red/far-red estimates to evaluate replicability and reproducibility of different needle arrangements in OC and FW. The SD of replicates within one arrangement configuration and method (n = 4–5) was used as an indicator of replicability. In turn, the SD of replicates for each arrangement configuration pooled over the two methods (n = 8–10) was used as an indicator to evaluate the reproducibility of the 1N, 3N, and NM arrangements.

## 3. Results

### 3.1. Baseline Correction

A simple baseline correction method Equation (1) was applied to each of the measured ChlF spectra (examples in Figure 4). Predictably, the baseline correction had a larger effect on the red ChlF tail wavelengths, where the filter transmittance and possible spectrometer resolution effects discussed in Section 2.3 were expected to have a larger impact. Importantly, the fact that REF3, a reference measurement acquired under the same conditions but with a Spectralon®panel instead of a leaf sample (Figure 4), presented non-zero values across the ChlF spectral range denoted that the baseline correction affected all ChlF wavelengths. The magnitude of the correction varied, however, from case to case, and from method to method, in part due to differences in leaf reflectance among samples (Supplementary Material, Figure S5). Leaf reflectance tended to be higher in the OC compared to FW and IS methods for birch and lingonberry, something expected due to inclusion of the specular reflectance component in OC, and which translated into an increased baseline correction. Interestingly, this was not the case for pine needles, which presented larger reflectance in FW than in OC and especially in IS. The performance of the baseline correction was also demonstrated in birch leaves illuminated with light sources with contrasting emission spectra: a halogen light (with substantial emission in the NIR) and a white LED light (with very small emission beyond 750 nm (see Supplementary Material, Figure S4)). Although the correction effect near the cut-off filter region was larger for the halogen light source compared to the LED, similar levels of correction were found across the rest of the ChlF spectral range, providing direct evidence for the saturation effect (I) described in Section 2.3. Baseline correction served also to corroborate that the black tape used as background in the OC method was indeed non-fluorescent for the range 680–850 nm (Supplementary Material, Figure S1). However, the remains of the uncorrected signal around the filter cut-off area reflect that the baseline correction method presented here still leaves room for improvement.

After correction, ChlF spectra were close to zero above 850nm in OC and FW, but this was not the case for the IS method, where the baseline correction clearly undercorrected the spectra throughout (Figure 4). It is important to note also that the slight increase in REF3 with wavelength observed across methods was not present in the raw data (digital counts) and was a result of the radiometric calibration function. Unless stated otherwise, all plots presented hereafter represent baseline-corrected ChlF spectra.

### 3.2. Comparison of Measurement Protocols

Despite purposeful adjustments to at-leaf surface PAR levels to similar levels across methods to minimize NPQ and to ensure that the resulting ChlF spectra would be readily normalized by PAR (Table 1), the resulting ChlF spectra still varied considerably among methods. For broadleaf (Figure 5a and Supplementary Material, Figure S6a) and single needle arrangements (i.e., 1N and 3N in Figure 6a–c), the OC delivered the highest average levels, whereas for pine mats the highest average levels were delivered by FW (Figure 6d–f). The IS method consistently yielded the smallest ChlF values across species, which was expected due to signal attenuation inside the sphere.

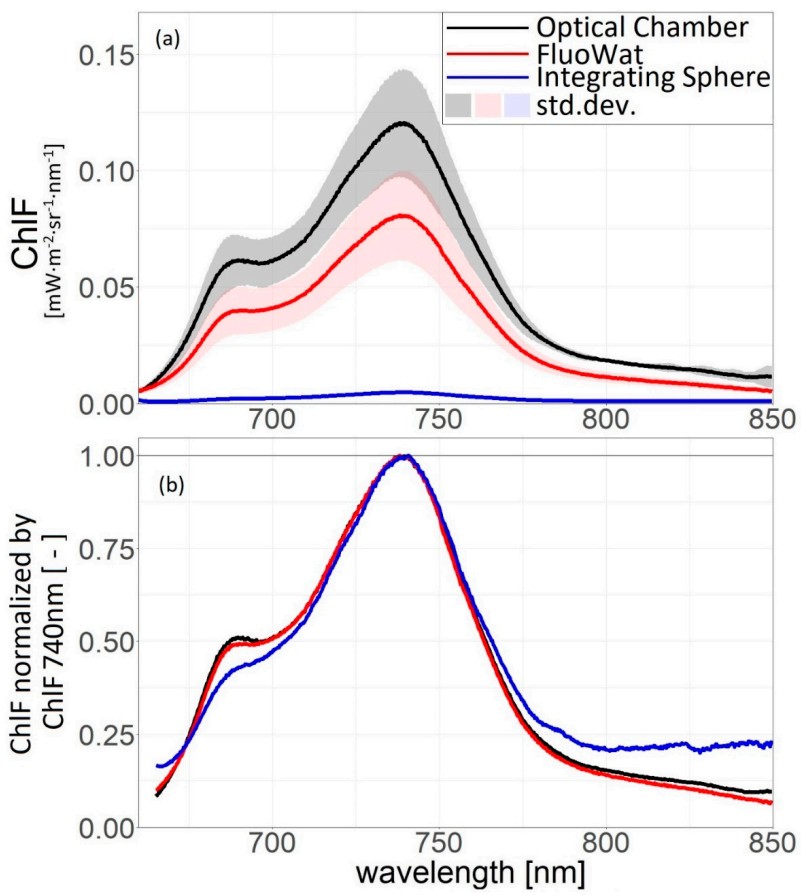

**Figure 5.** Chlorophyll fluorescence spectra of lingonberry in optical chamber (black), FluoWat (red), and integrating sphere (blue) methods. (**a**) measured spectra, standard deviation (n = 4–5) is marked as shading; (**b**) spectra normalized to F740.

We then normalized the ChlF spectra to the far-red peak at 740 nm, in order to examine differences in spectral shape (Figure 5b, Figure 6g–h and Supplementary material Figure S6b). The results indicated that the shape of ChlF spectra was quasi-identical when measured with the OC and FW methods, but presented a lower red peak and a broader far-red shoulder when measured in the IS, with red/far-red ratios significantly smaller for the IS compared to OC [p = 0.04 (lingonberry)] and FW [p = 0.02(lingonberry) and p = 0.01 (birch)]. Note that similarly to Figure 4e–f, the normalized spectra of the IS method also presented a substantial baseline beyond 800 nm.

### 3.3. Comparison of Needle Arrangements

As expected, the intensity of the recorded ChlF spectra increased with the number of needles both in the OC and FW setups, with smaller amplitude for the 1N arrangement, intermediate amplitude for the 3N, and largest for the needle mats (Figures 6a–c and 6d–f respectively). Again, we then

normalized the ChlF spectra to the far-red peak at 740 nm in order to examine the impact of needle arrangement on the shape of the ChlF spectra (Figure 6g,h). We found a general decreasing trend (not significant) in the normalized far-red peak (or the red/far-red ratio) with increasing number of needles in both OC and FW methods, possibly due to enhanced re-absorption.

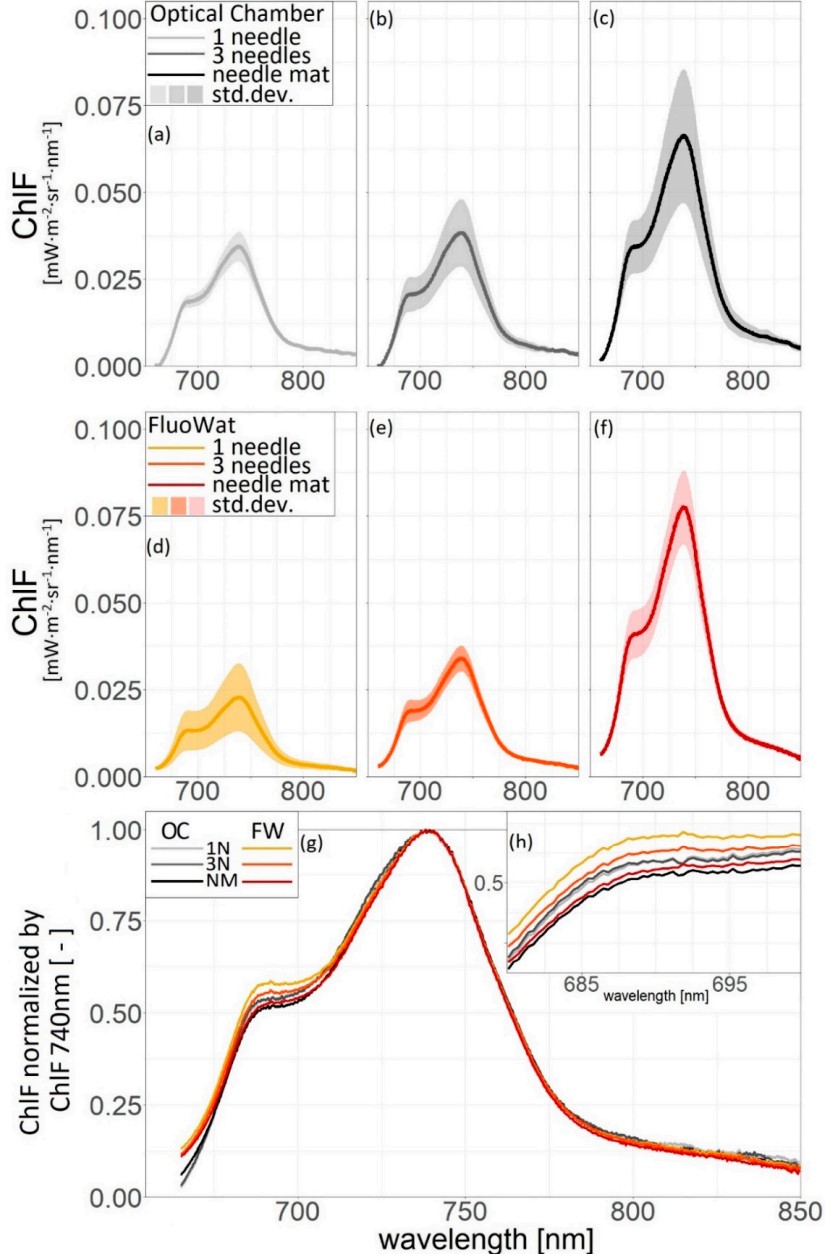

**Figure 6.** Chlorophyll fluorescence spectra of pine needles in the optical chamber (black shades) and FluoWat (yellow, orange, and red) methods. Top panel: measured spectra in the optical chamber: (**a**) 1N, (**b**) 3N, (**c**) NM, standard deviation marked as shading. Middle panel: measured spectra in FluoWat: (**d**) 1N, (**e**) 3N, (**f**) NM, standard deviation marked as shading; Bottom panel: both methods normalized to F740 (**g**) and zoomed in at the 675–700 nm range (**h**).

The replicability and reproducibility of measurements was evaluated using the SD of the estimated red/far-red ratios across replicates and needle arrangements, respectively. In both methods, NM showed highest replicability (i.e., the smallest SD) as compared to other needle arrangements—OC: 1N = 0.516 ± 0.045, 3N = 0.498 ± 0.036, and NM = 0.472 ± 0.018;

FW: 1N = 0.560 ± 0.06, 3N = 0.527 ± 0.055, and NM = 0.494 ± 0.019 (red/far-red ratio ± SD, n = 4–5). Similarly, NM also showed higher reproducibility across methods compared to 1N and 3N arrangements— 1N = 0.538 ± 0.055, 3N = 0.509 ± 0.048, and NM = 0.483 ± 0.021 (red/far-red ratio ± SD, n = 8–10).

The potential interference of the black tape used to hold the needles in place during the OC measurements could be ruled out since the tape did not register any substantial ChlF (Supplementary Material, Figure S1) and similar results were obtained for both the OC and FW when a light trap was used as the background.

### 3.4. Comparison of Leaf Morphology

We compared the shapes of the measured ChlF spectra across species for the OC and FW methods. For practical purposes, we present the results as normalized spectra, such that the level of the red region (685 nm) illustrates the red/far-red ratio. Only a very small effect of leaf morphology or sample arrangements could be seen in the far-red region of the spectra. Conversely, there were larger changes between methods in the red region of the spectra. Therefore, in addition to the various factors which control the intensity of the ChlF signal, differences in the red/far-red ratio highlight the presence of factors that affect the spectral shape of the ChlF.

Lingonberry (green) presented the lowest red/far-red ratio, with similarly low values in both OC (Figure 7a) and FW (Figure 7b). The one needle (light grey) and three needles (dark grey) configurations tended to yield systematically higher red/far-red ratios than NM (black) in both methods. Regarding red/far-red ratios, no statistically significant differences were found between the OC and FW method.

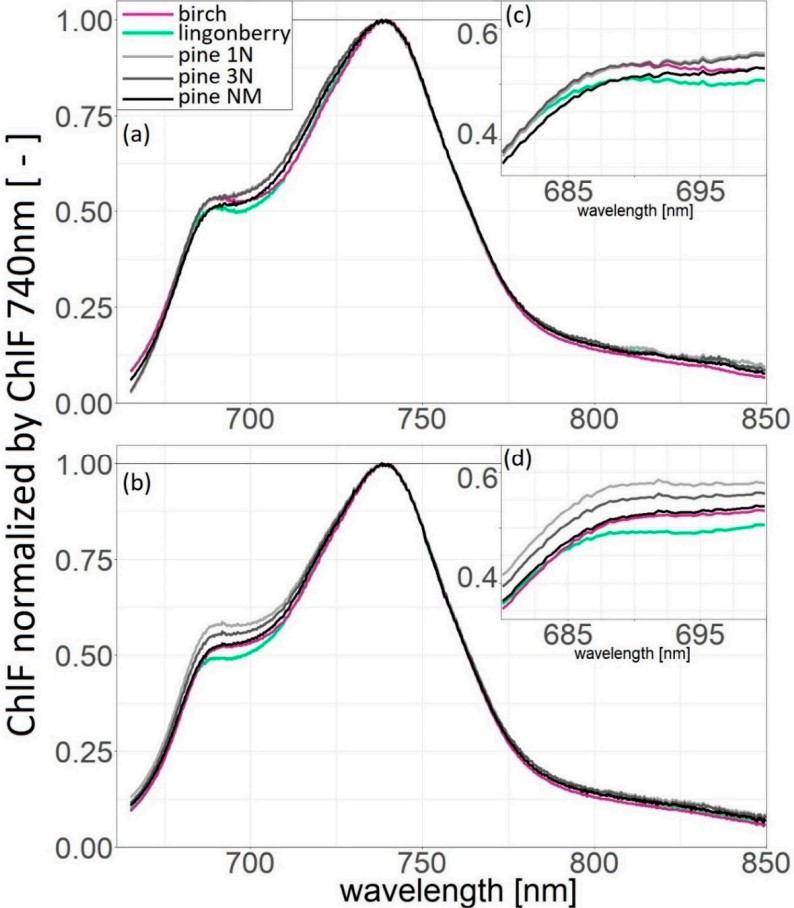

**Figure 7.** Chlorophyll fluorescence spectra of birch (violet), lingonberry (green) and pine needles (1 needle—light grey, three needles—dark grey, and needle mat—black): (**a**) in optical chamber (and zoomed in at the 675–700 nm range (**c**)); and (**b**) in FluoWat (and zoomed in at the 675–700 nm range(**d**)).

## 4. Discussion

### 4.1. Baseline Correction and the Red/Far-Red Ratio

Under certain conditions, such as when measuring in the direction of the specular reflectance, applying long ITs to boost SNR in the ChlF spectral range at the expense of signal saturation, or using spectrometers with limited dynamic ranges or cut-off filters with substantial transmissions (OD > 4), the resulting ChlF spectra may be contaminated with an undesired and wavelength-dependent baseline component. Although the strength and consequences of this overestimation may vary from case to case, baseline correction can be critical when comparing datasets acquired with different setups or species. Baseline correction is especially useful in recovering the spectra near the filter cut-off. As shown (Figure 4 and Supplementary Material, Figure S4a), the red tail is the region where the correction is strongest. Since the red ChlF peak is closer to the filter cut-off region than the far-red peak, the baseline effect could potentially interfere with the relative heights of the ChlF peaks and, thus, the red/far-red ratio [18].

Red to far-red ChlF ratios have been used as indicators of [Chl] [41,42] and to track physiological stress responses [15,17,43,44]. Variations in red to far-red ChlF have been also used to estimate the relative contribution of PSI and PSII to the ChlF signal [45,46]. It is, therefore, important to ensure that the baseline effect, when present, is properly addressed. Comparison of measured and corrected spectra showed that baseline application changed the red/far-red ratio by only 2–3% (OC) and by 1.1–2.1% (FW). The larger baseline effect in the OC compared to the FW could be explained in terms of enhanced reflectance in the OC where the specular reflectance component is also contributing to the reflected signal (Supplementary Material, Figure S5). Overall, despite the limited impact of the baseline correction method on the shape of the resulting ChlF spectra, we recommend taking into account the possibility of a significant baseline during data interpretation. Using spectrometers with a high dynamic range so as not to saturate the signal, and optical filters with an OD equal to or larger than 4, is also recommended. When applying this baseline correction method, we also recommend measuring an REF3 for each sample rather than for each setup as we did in the present study, since filter repositioning from sample to sample might cause some variation in the REF3, which is probably one of the reasons why the correction of the black tape ChlF spectra (Figure S1) presented still some uncorrected "photons" below 680 nm.

Finally, the baseline correction did not work properly for the IS method where substantial levels of "ChlF" were still estimated at 850 nm and beyond. This inconsistency could be attributed to the small magnitude in the signal relative to the background noise compared to the other methods. Although measurements at higher light intensities where outside the scope of the present study, they could serve to test and confirm the validity of the baseline correction for the IS setup.

### 4.2. Measurement Protocols

The most pronounced differences among the three tested methods were in the amplitude of the recorded ChlF signal: highest for OC (in birch and lingonberry) and FW (in pine), and smallest for IS (Figures 5a and 6a–f), despite having similar PAR levels at the leaf surface (Table 1). These differences can be attributed to contrasting geometries between setups—measurements were acquired at nadir in the backscatter direction for OC, at a 45° viewing angle respective to the illumination beam in FW, and integrated across all directions in the IS method. Interestingly, the signal was largest in the OC for broadleaf species, but not for pine needle mats, for which the FW presented larger values. We are not certain about the underlying reason for this is, but suggest it may reflect a decrease in overall absorbed PAR (APAR) when illuminating the needles at nadir, relative to the FW, due to the effects of the 3D structure of the needle mats and small gaps in between. In other words, the apparent gap fraction of needle mats tends to be smaller when viewed at 45° compared to nadir, resulting in lower PAR loses through the gaps. Bidirectional ChlF measurements of needle mats under controlled laboratory conditions would be required to further investigate these differences. In turn, signal attenuation

inside the sphere would explain why the amplitude of the ChlF signal was smallest in the IS method compared to others.

The spectral shape we recorded was consistent with examples from literature measured under both ambient and artificial illumination, using both halogen and LED light sources. Van Wittenberghe et al. (2013) measured solar-induced leaf-level upward ChlF with FluoWat method. The results for two broadleaf species (see Figure 7 in [40], red/far-red ratio = 0.6) represent similar spectral distributions as our birch and lingonberry in both OC and FW. Corresponding red/far-red ratio values (0.48–0.62) were also measured for Sun leaves of birch and pine by Atherton et al., while measured with FW and white halogen illumination [38]. Similar spectra with red/far-red ratios below unity have been also recorded in response to red LED illumination measured with a high-resolution QePro spectrometer coupled to an IRGA system [47]. Although assessing the effect of light quality was outside the scope of the present study, we found highly comparable ChlF shapes for birch leaves illuminated with a white LED and halogen light (see Supplementary Material, Figure S4b), suggesting that small changes in light quality would not cause a drastic change in spectral shape. In contrast, highly different red/far-red ratios have been reported when comparing leaves illuminated with blue vs. red LED light illumination [47]. It is also important to remember that our measurement were conducted under low light conditions, given the requirement of minimum NPQ and, therefore, the red/far-red ratios might be slightly overestimated compared to leaves measured under field conditions with strong light.

After normalizing to F740 nm (Figure 5b, Figure 6g,h and Supplementary Material, Figure S6b) the shapes of the spectra obtained with OC and FW were generally very consistent with each other suggesting that differences in measuring geometry would not appear to affect the shape of the resulting ChlF spectra. Together with the mentioned relation between LED and halogen induced ChlF, this finding could have important implications for future leaf-level ChlF meta-analysis studies. Potentially, data from different measuring setups could be combined together to study how physiological, structural, and biochemical factors control the shape of the leaf-level ChlF spectra.

The similarity between observations from two contrasting viewing angles (nadir for OC and 45° for FW), would indicate a certain degree of isotropy in the shape of ChlF spectra. In contrast, IS delivered a substantially smaller far-red peak, as compared with OC and FW. In addition, the baseline correction method did not work properly for the IS method, thus its effect on the red/far-red ChlF cannot be ruled out. However, we believe that the observed decrease in red/far-red ratio inside the sphere is genuine and could be explained in terms of enhanced red ChlF re-absorption inside the sphere [18,42], where scattered red photons are re-absorbed by the leaf sample itself before reaching the detector.

Although measurement of ChlF properties using IS could prove to be useful—for estimating ChlF spectral properties of leaves with a complex geometry or even of small shoots, by placing the sample inside the sphere [33]—the enhanced re-absorption of red ChlF inside the sphere might be a limitation. Tests at higher light intensities, with the purpose of increasing SNR and improving the performance of the baseline correction, would be needed to assess the informative potential of the IS method for measuring ChlF spectral properties.

*4.3. Needle Arrangements and Morphology Effect*

We compared three alternatives to measuring ChlF spectra in needle-like leaves, measuring: a single needle (1N), needles separated by gaps (3N), and a full needle mat with minimal gaps (NM). We found that NM yielded higher replicability in the resulting shape of the spectra, as indicated by lowest SD in both OC and FW methods. The results showed that the NM presented higher replicability and reproducibility across methods compared to the 1N and 3N arrangements, indicating that this might be a desirable arrangement for studies intended to track temporal changes in ChlF properties for needle-like species. We are aware that arranging needles in mats (see Section 2.1.) could potentially enhance self-shading, multiple scattering across all ChlF wavelengths, as well as re-absorption of predominantly red wavelengths by adjacent leaves [27,31]. With a single needle, ChlF photons emitted

in lateral directions will not reach the sensor. With a needle mat, ChlF photons emitted in lateral directions can still be scattered by adjacent needles in the direction of the sensor and contribute to the measured ChlF. Since red ChlF photons have a higher probability of being re-absorbed by adjacent needles, compared to far-red photons, one would expect the laterally-scattered ChlF contribution to be enriched in far-red photons, reducing the overall red/far-red ratio. This phenomenon could explain why the red/far-red ratios of NM were smaller than those of 1N and 3N arrangements across methods (Figure 6g,h). Interestingly, assuming that the 1N arrangement would have provided a more realistic estimate of the red/far-red ratio, it is still controversial why needles presented higher red/far-red ratios compared to birch or lingonberry (Figure 7b). Two explanations can be excluded in our study. First, because all the measurements were purposefully conducted at low PAR, any differences between leaves in terms of NPQ should be minimal. Second, because the leaves showed $F_V/F_M$ levels typical for the species in the absence of stress (see Section 2.1.), the potential impact of sustained physiological stress can also be ruled out [11,38,39]. We, therefore, suggest that the slightly higher levels of red/far-red ChlF ratio observed in 1N and 3N relative to those observed in birch and lingonberry leaves would respond to differences in [Chl] induced reabsorption coupled with differences in leaf morphology.

One important question is raised from this observation: does the red/far-red ChlF ratio of leaves relate better to the total [Chl] expressed on a surface basis or on a leaf mass basis (possibly better related to the density of chlorophyll in the leaf)? Twice as high levels of [Chl] expressed on a surface basis have been observed for boreal Scots pine compared to boreal silver birch [38] during summer, and since higher [Chl] should increase reabsorption of red ChlF, the result does not appear to be consistent with our observation of higher red/far-red ratio in pine compared to birch. In contrast, when [Chl] is expressed on per leaf mass basis, almost twice as high levels of [Chl]/g DW are reported for boreal silver birch compared to Scots pine [48] during summer. This finding suggests that red/far-red ratio might be more closely linked to [Chl]/g DW than [Chl]/$cm^2$, and that not only leaf-level [Chl] but also morphology could play a role in shaping the ChlF spectra.

The decrease in red/far-red ratio observed for NM, as compared to 1N and 3N arrangements, indicates that arranging the needles may to some extent underestimate the ratio and it should be considered when quantitatively comparing ChlF spectra from needles and broadleaf species. This observation emphasizes the complexity of dealing with $G_F$ in needle-like leaves. As widely suggested in the literature [27,29,32], gaps are both inevitable and a source of considerable error when measuring R and T of needle like leaves. In the case of ChlF, accurate measurements are further complicated by the phenomena of re-absorption.

Further work to investigate to what extent needle mats underestimate the red ChlF and whether new solutions would be possible to measure single needles without compromising the strength of the signal or the replicability of the measurement is required. Finally, it is also important to remember that needle mats—despite being relatively easy to measure – are not a realistic representation of the natural geometry of collections of needle leaves, commonly referred to as shoots. Hence, each scale of the ecosystem—from individual leaves and shoots to individual trees and plants to the forest canopies—has their own specific and interacting set of optical properties that combine to influence the observed remote sensing retrieval [9,10]. Measurements and knowledge of each scale is, therefore, required to accurately infer physiology from coarse resolution remote measurements.

## 5. Conclusions

The interpretation of solar-induced fluorescence (SIF) acquired from towers, drones, aircrafts, or satellites, depends largely on our capacity to characterize and model the impact that different factors exert on the observed results. Variability in amplitude and shape of the leaf-level ChlF spectra appears across species, stress factors, and canopy light gradients. In order to link signals acquired at different scales, we need to develop standardized protocols to measure leaf-level ChlF properties across different types of leaves. In this study, we measured leaf-level ChlF spectra using methods representing notable differences in measurement geometry. We tested three species and

compared the impact of needle arrangement on the resulting ChlF spectra. We found that the shape (but not the amplitude) of the leaf-level ChlF spectra was very robust in response to changes in illumination/viewing geometry across OC and FW methods. The IS method, despite being a standard for reflectance and transmittance measurements, did not appear to be a good solution to measure ChlF spectra, as it tended to underestimate the red ChlF wavelengths. We also presented a simple method to correct for baseline effects and emphasized its potential importance under certain conditions. Furthermore, we recommend the use of needle mats—which presented good replicability across FW and OC—as a reasonable compromise and reproducible method to measure temporal changes in ChlF spectra of needle-like leaves until better solutions become available and keeping in mind the potential underestimation of red ChlF due to re-absorption.

The ideal protocol is always the one that best adjusts to its purpose and application. The FW method makes it possible to measure leaf ChlF properties both in situ as well as in the laboratory, where incoming PAR can be standardized for easier comparisons. The OC method instead provides the possibility of measuring leaf ChlF properties at both ambient and 77 K temperatures, but cannot be applied to leaves in situ. Other methods can be developed to serve still other purposes. The key point is to keep measurement setups consistent across experiments, as well as thoroughly documented for potential future use in meta-analysis, so that ChlF variation can be attributed to biochemical, physiological, and structural factors.

**Supplementary Materials:** Supplementary materials are available online at http://www.mdpi.com/2072-4292/11/5/532/s1.

**Author Contributions:** Conceptualization, P.A.R. and A.P.-C.; Data curation, P.A.R.; Formal analysis, P.A.R.; Funding acquisition, A.P.-C.; Investigation, P.A.R.; Methodology, P.A.R., J.A., L.A. and A.P.-C.; Supervision, J.A. and A.P.-C.; Writing – original draft, P.A.R. and A.P.-C.; Writing – review & editing, P.A.R., J.A., L.A. and A.P.-C.

**Funding:** This research was supported by Academy of Finland FLUOSYNTHESIS project #293443 and #288039. Paulina A. Rajewicz acknowledges the Doctoral Programme in Atmospheric Sciences (ATM-DP, University of Helsinki) for financial support. Luis Alonso is currently partially funded by AVANFLEX project (Advanced Products for the FLEX mission), n° ESP2016-79503-C2-1-P, Ministry of Economy and Competitiveness, Spain.

**Acknowledgments:** We would like to thank Markus J. Haapala, from the Division of Pharmaceutical Chemistry and Technology, Faculty of Pharmacy, University of Helsinki, Finland for assistance with the 3D printing of the optical chamber, and Alyce M. Whipp from University of Helsinki Language Services for help with language revision. We would also like to express our gratitude to the anonymous reviewers for their detailed and constructive comments which significantly improved our manuscript.

**Conflicts of Interest:** The authors declare no conflict of interest.

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
