# Peer review of "Leaf-Level Spectral Fluorescence Measurements: Comparing Methodologies for Broadleaves and Needles"

_remotesensing, doi:10.3390/rs11050532_

Round 1
Reviewer 1 Report
I have read through the responses to all reviewer comments on this paper, and I think the authors did a nice job addressing them. In some cases, I found comments from other reviewers in round 1 to be unreasonable, yet the authors cleverly navigated these issues.
Following revisions, and after seeing the response of one of the reviewers to the revisions (who still suggests major revisions), I would like to suggest that these comments are out of line. The comments from the aforementioned reviewer are too broad to be adequately addressed at this stage. And I also disagree with them – they seem to be emotionally charged, and not grounded in reason. I am of the opinion that the authors present a clear, robust study that is important for advancing leaf level ChlF measurements into the future. I commend the authors on a great job and look forward to seeing this in press. I only have some very minor things to consider below:
I’m not sure the last sentence of the abstract is necessary. While the results suggest increased red absorption when needles are placed in the ‘needle-mat’ formation, I’m not sure that is exactly what is happening. This might make sense if the needles were thicker, or had more Chl. But since the penetration depth in the incident light direction doesn’t actually change I’m not sure this statement can be made directly.
Line 47 – technically (a) and (c) here are intrinsically related as we assume that any stress is actually modulating the relative contribution of PSI and PSII ChlF. And while Chl concentration is implied in (b), it might be worth mentioning explicitly.
Line51-53- not necessarily by the dynamic response of ‘leaf-level photosynthesis to stress’ but you could just say ‘response of leaves to stress’. Also, isn’t this sentence a bit redundant with the first sentence of the paragraph? As is the one on lines 54-55. This paragraph could use some cleaning up.
Line 63 – ‘irrespective of...’
Line 89 – this sentence is not complete.
Lines 91-95 – the word ‘level’ is a bit confusing. Do you mean scale? Or canopy position?
I appreciate the additional introduction of Gap Fraction on 86-95.
Fig. 1 – the approximate FOV is a nice addition.
Line 187 – ‘off’ not ‘of’
Line 191- is there any data to report anyhow? I think you could remove ‘data not reported’.
Fig. 3 is very nice and helps the reader immensely.
Line 312 – ‘attempt’
The explanation of the ‘baseline’ correction seems to be improved in this version and I appreciate the changes made to Fig. 4.
Line 418 – ‘no’ statistically significant differences…
Line 437 – aren’t the ‘stress responses’ directly manifested in PSI/PSII contributions anyhow?
A thought: The incident PAR level is very low (50 and 38 umol). It seems to me that if the light intensity was stronger the SNR would improve, is this worth discussing? I’ll leave it up to the authors.
Line 526-535 – this is an important question! I would make it its own paragraph. Would be nice to have pigment measurements for this study…but maybe future studies will include those.
Author Response
Thank you for your comments regarding our re-submitted manuscript. Our response can be found in the attached Word file.
Paulina Rajewicz, corresponding author

Reviewer 2 Report
I think this manuscript has been well improved according to the reviewers' comments. It well be a good reference for SiF measuring.
Author Response
We would like to sincerely thank you for your comments regarding our re-submitted manuscript. We are happy that the changes we have made to the first version of our manuscript met your expectations well. We appreciate that you find our revised manuscript valuable enough to be accepted for Remote Sensing Journal.
Reviewer 3 Report
Dear Authors,
there a lot of similar analyses and measurements. You have to present it explaining what do you propose new: methods, ideas, results? So, please, could you present deeper a theoretical background of your topic. This same you need to improve the Discussion presenting your achievements and their comparison to results of other researchers.
Methods should be better described. I propose to prepare a research schema showing all steps, and then to describe all your data acquisition and processing.
My activity is similar, so I tried to follow your methods, and it wasn't easy task, there are a lot of gaps between data acquisition and processing.
You should be more detailed, e.g. in error assessment and validation of your methods.
Much more details you can find in the attached manuscript.
Best regards
Reviewer

Author Response
Thank you for your comments regarding our re-submitted manuscript. Our responses can be found in the attached Word file.
Paulina Rajewicz, corresponding author

Round 2
Reviewer 3 Report
Dear Authors,
thank you for your revision. The manuscript looks better, but there some improvements are still needed.
In the Introduction you have a nice theoretical background, but your should present current solutions as well. The Methods need more verification and error assessment methods. Please, look at the graphs in the Results, standard deviations aren't visible. You have to describe the accuracies.
The Discussion needs much more direct comparisons of your result with achievements of other researchers to know what kind of methods offer better results.
Please, look at the figures, the quality is low, and font size aren't proper, please look at the guidelines. You need to add your interpretation of all figures presented in Results. Some graphs (more important) could be moved from the supplement. There are too many graphs.
Much more comments you can find in the attached manuscript.
best regards
Reviewer

Author Response
We would like to thank you for your comments regarding our re-submitted manuscript. Please find our response to your comments in the attached file.
Sincerely, Paulina Rajewicz - corresponding author
